# Characteristics of Bacterial Community Structure and Function in Artificial Soil Prepared Using Red Mud and Phosphogypsum

**DOI:** 10.3390/microorganisms12091886

**Published:** 2024-09-13

**Authors:** Yong Liu, Zhi Yang, Lishuai Zhang, Hefeng Wan, Fang Deng, Zhiqiang Zhao, Jingfu Wang

**Affiliations:** 1College of Biological and Environmental Engineering, Guiyang University, Guiyang 550005, China; 2Guizhou Institute of Biology, Guiyang 550009, China; 3State Key Laboratory of Environmental Geochemistry, Institute of Geochemistry, Chinese Academy of Sciences (IGCAS), Guiyang 550081, China

**Keywords:** bacterial community, microbial function, artificial soil, red mud, phosphogypsum, soil evolution

## Abstract

The preparation of artificial soil is a potential cooperative resource utilization scheme for red mud and phosphogypsum on a large scale, with a low cost and simple operation. The characteristics of the bacterial community structure and function in three artificial soils were systematically studied for the first time. Relatively rich bacterial communities were formed in the artificial soils, with relatively high abundances of bacterial phyla (e.g., *Cyanobacteria*, *Proteobacteria*, *Actinobacteriota*, and *Chloroflexi*) and bacterial genera (e.g., *Microcoleus_PCC-7113*, *Rheinheimera*, and *Egicoccus*), which can play key roles in various nutrient transformations, resistance to saline–alkali stress and pollutant toxicity, the enhancement of various soil enzyme activities, and the ecosystem construction of artificial soil. There were diverse bacterial functions (e.g., photoautotrophy, chemoheterotrophy, aromatic compound degradation, fermentation, nitrate reduction, cellulolysis, nitrogen fixation, etc.), indicating the possibility of various bacteria-dominated biochemical reactions in the artificial soil, which can significantly enrich the nutrient cycling and energy flow and enhance the fertility of the artificial soil and the activity of the soil life. The bacterial communities in the different artificial soils were generally correlated with major physicochemical factors (e.g., pH, OM, TN, AN, and AP), as well as enzyme activity factors (e.g., S-UE, S-SC, S-AKP, S-CAT, and S-AP), which comprehensively illustrates the complexity of the interaction between bacterial communities and environmental factors in artificial soils, and which may affect the succession direction of bacterial communities, the quality of the artificial soil environment, and the speed and direction of the development and maturity of the artificial soil. This study provides an important scientific basis for the synergistic soilization of two typical industrial solid wastes, red mud and phosphogypsum, specifically for the microbial mechanism, for the further evolution and development of artificial soil prepared using red mud and phosphogypsum.

## 1. Introduction

Red mud is a strongly alkaline solid waste produced by the alumina industry, and phosphogypsum is a strongly acidic solid waste produced by the phosphorus chemical industry. As two typical bulk industrial solid wastes, red mud and phosphogypsum are both produced in large quantities: about 1~2 t of red mud is produced for every 1 t of alumina and about 4~5 t of phosphogypsum is produced for every 1 t of phosphoric acid. The global cumulative production of red mud and phosphogypsum has reached 4 billion tons and 7 billion tons, respectively, and is still growing, with 120 million tons and 250 million tons per year, respectively [1,2,3]. For a long time, due to the lack of effective disposal technology, red mud and phosphogypsum have often been stored. Red mud and phosphogypsum are highly corrosive and contain harmful elements, such as heavy metals, which cause huge potential risks to the surrounding environment and human health [2,4,5]. For example, red mud and phosphogypsum dust particles easily cause atmospheric particulate pollution, which leads to corrosion hazards and heavy metal toxicity in the human body, animal respiratory tracts, skin, and plant leaves. Red mud and phosphogypsum leach into rainwater, and the leachate produced contains a variety of pollutants, which can easily pollute the soil and water bodies in nearby areas with heavy metals, phosphorus, and fluorine, as well as runoff, thereby causing long-term toxic risks to the food crops and drinking water for surrounding residents [3,6]. Similarly, the strong acid–base corrosion of leachate and phosphogypsum also causes obvious acid–base changes in soil and water bodies. It can significantly affect the normal growth of plants and aquatic organisms, and it can even cause a large-scale death phenomenon. Moreover, red mud and phosphogypsum are often piled up in mountains, which easily leads to the occupation of large areas of land resources. Furthermore, with the increasing number and accumulation of storage yards, the related maintenance costs and management risks are significantly increased, and the slightest carelessness may result in dam breakage, producing immeasurable serious consequences. In summary, the mass production and accumulation of red mud and phosphogypsum have caused a series of risks and challenges in the normal production of relevant enterprises and the stable development of industry, the local economy, social governance, and government management.

In recent years, the resource utilization of red mud and phosphogypsum has become a wide concern and more research has been carried out. Related technologies are also being developed. For example, red mud and phosphogypsum are used in construction materials (such as in bricks, roadbeds, cement, gypsum board, and ceramics) [7,8] and soil amendments (red mud can passivate soil heavy metals and reduce soil heavy metal activity; phosphogypsum can acidify and improve saline–alkali soil and prevent soil compaction) [9,10]. Red mud and phosphogypsum can be good adsorption materials (absorbing carbon dioxide, heavy metals, organic matter, etc.) after different modifications [11,12,13]. In addition, red mud and phosphogypsum are rich in iron, aluminum, calcium, phosphorus, fluorine, heavy metals, rare earth elements, etc., the recovery values of which are high, and some new technologies can be used to recover these elements as resources [14,15,16]. Although the research and technical applications of the resource utilization of red mud and phosphogypsum show a significant increasing trend, so far, the resource utilization of the two is still mainly based on theoretical research, many technologies are not mature (with shortcomings, instability, and limitations), industrial applications are still few, and the reduction of the two is still very limited in general. It does not change the fundamental problem that the annual reduction is far less than the amount of production. Therefore, the resource utilization technology of red mud and phosphogypsum, especially large-scale and low-cost technology, has become a research hotspot. In the early stage, our research team prepared an artificial soil based on the strong acid–base neutralization reaction of red mud and phosphogypsum, which contain many beneficial elements of soil (O, Si, Fe, Al, Ca, K, Mg, S, P, etc.), and the clay mineral properties of red mud itself, supplemented by other materials. It has been proven that artificial soil technology has great potential and broad market prospects for realizing the large-scale utilization of red mud and phosphogypsum [17].

However, artificial soil is still in the initial stage of the synergistic soilization of red mud and phosphogypsum. Although it has chemical element compositions (O, Si, Al, Fe, Na, Mg, etc.) similar to those in common natural soils, common physical and chemical characteristics, and major nutrients (nitrogen, phosphorus, and potassium), and it can be adapted for the growth of some plants, etc., the continuous development of artificial soil will still take a long time and a continuous evolution process is required for it to mature into real soil. Bacteria are the most abundant biological groups in the soil ecosystem, the main participants in soil biogeochemical processes, and the key drivers of soil biogeochemical processes such as oxidation, nitrification, ammonization, carbon fixation, nitrogen fixation, and vulcanization [18]. Bacteria and the soil environment are very sensitive and interact with each other [19]. On the one hand, the structure of bacterial communities effectively indicates the environmental quality, nutrient fertility, material circulation, and other information about the soil [20,21]. In particular, bacteria are the key contributors to the formation and cultivation of soil fertility and its continuous development and maturation, and to the continuous improvement in and enrichment of micro-ecosystems [22]. On the other hand, environmental factors, such as the pH, moisture, organic matter, and nutrients in the soil, are the basis for the survival of bacteria, and they significantly affect the growth and metabolism of bacteria and the diversity of bacterial communities, which, in turn, affect the direction of the evolution of the soil environmental quality [23]. Therefore, the survival and diversity of the bacterial community in artificial soil not only indicates its environmental quality but also determines its environmental quality evolution and its direction and speed of further development and maturity. Based on our previous research ideas and schemes [17], the communities and their functional characteristics of three different artificial soils prepared using red mud and phosphogypsum were studied. This study provides an important scientific basis for a further understanding of the microbiological mechanisms of the further evolution and development of red mud- and phosphogypsum-based artificial soil.

## 2. Materials and Methods

### 2.1. Preparation of Artificial Soil

The main materials, red mud and phosphogypsum, were obtained from a red mud yard and a phosphogypsum yard in Guizhou, China, with pH values of 11.4 and 1.8 and moisture contents (W_H2O_) of 23.1% and 21.9%, respectively. The auxiliary materials included common rice husk powder, bentonite, fly ash, and polyacrylamide flocculant. Red mud and phosphogypsum were naturally dried and ground through a 40-mesh sieve and mixed evenly according to a mass ratio of 2.5:1 to form a neutral red mud–phosphogypsum substrate (200 g). Then, the above four auxiliary materials were added, and the mass compositions are as described in Table 1. After full mixing, 10 mL of a dilute suspension of microorganisms was added (mainly bacillus, e.g., *Bacillus amyloliquefaciens*, *Bacillus licheniformis*, *Bacillus subtilis*, with an effective viable bacterial count > 0.4 million/mL) to form three kinds of treated artificial soils, namely, T7, T8, and T9.

### 2.2. Sample Collection and Determination

After the three artificial soils were prepared as described in Table 1, they were placed in pots, and artificial watering was employed to retain about 30% of the soil moisture. Three batches of plants (a total of 12 plants, such as rapeseed, chili pepper, alfalfa, and rye grass) were used, seed germination and seedling growth tests were conducted successively in a greenhouse (temperature: about 25~30 °C), and the common physicochemical indexes, main nutrients, and different enzyme activities of the artificial soils were analyzed regularly, as previously described in reference [17]. After 1 year of the pot plant experiment, the artificial soil samples of T7~T9 were collected and further divided by freeze-drying, crushing, and foreign matter removal, and the quartered method was used for the analysis of the soil bacterial community, basic physicochemical indexes (pH, moisture content (W_H2O_), organic matter (OM), cation-exchange capacity (CEC)), main nutrients (total nitrogen (TN), total phosphorous (TP), total potassium (TK), available nitrogen (AN), available phosphorous (AP), and available potassium (AK)), and different enzyme activities (catalase (S-CAT), sucrase (S-SC), urease (S-UE), alkaline phosphatase (S-AKP), and acid phosphatase (S-AP)). All samples were analyzed in three parallel sets.

The bacterial communities in the artificial soil samples were analyzed using high-throughput sequencing by Shanghai Majorbio Bio-pharm Technology Co., Ltd., Shanghai, China (https://www.majorbio.com). The detection method for the sample DNA purity and concentration was NanoDrop2000, and the detection method for the DNA integrity was agarose gel electrophoresis (after melting the samples on ice, fully mixing, and centrifuging, an appropriate number of samples was taken for detection; the agarose gel concentration was 1%, the agarose gel voltage was 5 V/cm, and the time was 20 min). For the PCR experiment, the bacterial primer (338F_806R) design was 338F: ACTCCTACGGGAGGCAGCAG and 806R: GGACTACHVGGGTWWTCTAAT; the cycle number was 27; and the annealing temperature was 55 °C. The methods for analyzing the basic physicochemical indexes, main nutrients, and different enzyme activities of the three artificial soils were the same as those described in reference [17]; that is, they mainly referred to the following Chinese standards or common methods: W_H2O_ (weighing method); pH (soil–water mass ratio of 1:2.5); OM (GB 9834-88) [24]; CEC (HJ 889-2017) [25]; TN (LY/T 1228-2015) [26]; AN (DB51/T-1875-2014) [27]; TP (GB 9837-88) [28]; AP (NY/T 1848-2010) [29]; TK (LY/T 1234-2015) [30]; AK (NY/T 889-2004) [31]; different enzyme activities (mainly referring to reference [32]: S-SC (3,5-dinitrosalicylic acid colorimetric method); S-CAT (potassium permanganate titration method); S-AP/AKP (sodium phenyl phosphate colorimetric method); S-UE (phenol sodium-hypochlorite colorimetric method)).

### 2.3. Data Analysis

The data on the artificial soils are presented in the tables using the mean and standard deviation of parallel samples. Significant differences in the data were analyzed using single-factor ANOVA with LSD. All data on the microbial (bacterial and fungal) compositions in the artificial soils were analyzed on the online Majorbio Cloud Platform (www.majorbio.com).

## 3. Results and Analysis

### 3.1. Physicochemical Parameters, Main Nutrients, and Enzyme Activities of Artificial Soils

As described in Table 2, Table 3 and Table 4, the pH values of T7~T9 ranged from 8.4 to 8.7, and the value of T9 was significantly higher than those of T7 and T8 (*p* < 0.05). The three artificial soils could hold a certain amount of water, and the W_H2O_ was between 30.1% and 31.7% (Table 2). OM was abundant in the three artificial soils, and the proportion of OM was between 5.0 and 5.5%, with the value of T9 being significantly lower than those of T7 and T8 (*p* < 0.05). The CECs of the three artificial soils were close, ranging from 11.1 to 12.3 cmol/kg. The TN and AN contents in T8 were the highest, at 596.0 mg/kg and 211.4 mg/kg, respectively, which were significantly higher than those of T7 and T9. The TP contents in the three artificial soils were high and showed no significant difference, ranging from 2578.6 to 2842.9 mg/kg, but the AP contents in the three artificial soils were significantly different, ranking in the order of T8 > T7 > T9 (*p* < 0.05). There was no significant difference in the TK and AK contents among the three artificial soils. The TK contents were all high, ranging from 31.8 g/kg to 38.6 g/kg, while the AK contents were all low, ranging from 14.2 to 15.4 mg/kg. Certain enzyme activities were accumulated in the three artificial soils, and the S-CAT and S-AP were in the range of 1.121~1.285 mg/L and 0.028~0.029 mg/g, respectively, at 24 h. The S-AKP ranged from 0.329 to 0.561 mg/g at 24 h, with significant differences between the artificial soils, ranking in the order of T9 > T8 > T7 (*p* < 0.05); the S-UE in T9 was the highest (0.789 mg/g, 24 h), which was significantly higher than that in T7 and T8. In T8, S-SC was the highest (1.762 mg/g, 24 h), which was significantly higher than that in T7 and T8.

### 3.2. Alpha Diversity Index Analysis of Bacterial Communities in Artificial Soils

The alpha diversity index of microbial communities can be used to assess the richness, diversity, and uniformity of microorganisms in environmental samples. There were some differences in the alpha diversity indexes of the bacterial communities in the three artificial soils (Table 5). The Sobs indexes of the three artificial soils were significantly different, ranking T7 > T8 > T9 (*p* < 0.05). The Ace index and Chao index of T7 were similar to those of T8 but significantly higher than those of T9 (*p* < 0.05), indicating that T7 had relatively higher bacterial community richness; that is, its number of bacteria was the highest, followed by that of T8, and that of T9 was the lowest (i.e., it had the minimum number of bacteria). The Shannon index values of T7 and T9 were close to each other and significantly higher than that of T8 (*p* < 0.05), while there was no significant difference in the Simpson indexes of the three artificial soils (*p* > 0.05), indicating that the bacterial community diversities of the three artificial soils were relatively small, and the bacterial community diversities of T7 and T9 were slightly higher than that of T8. The Pielou_e index values of the T7 and T9 bacterial communities were close to each other and significantly higher than that of T8 (*p* < 0.05), indicating that T7 and T9 had better community evenness than T8; that is, different microbial species were distributed more evenly. The coverage indexes of the three artificial soils were greater than 0.99, indicating that almost all the bacteria in T7~T9 were detected, which well reflects the real situation of their bacterial community compositions.

### 3.3. Composition of Bacterial Communities in Artificial Soils

The Venn analysis of the bacterial communities in the three artificial soils showed that there were 749 identical OTUs (58.52%) in T7~T9, and 127, 69, and 66 unique OTUs in them, respectively (Figure 1). At the bacterial phylum level, there were 23 identical phyla in T7 and T8 (82.14%). T7 had a unique phylum, while T8 and T9 had no unique phyla. T7 and T8 had two identical phyla, and T8 and T9 had two identical phyla (Figure 1). At the bacterial genus level, there were 377 identical genera (70.07%) in T7~T9, and 51 (9.48%), 19 (3.53%), and 10 (1.86%) unique genera, respectively (Figure 1). At the phylum level (Figure 2), the bacterial phyla with abundances greater than 5% in T7 included *Cyanobacteria* (34.79%), *Proteobacteria* (16.05%), *Actinobacteriota* (14.28%), *Chloroflexi* (10.93%), and *Bacteroidota* (6.92%). The total was 82.97%. The bacterial phyla with abundances greater than 5% in T8 included *Cyanobacteria* (36.98%), *Proteobacteria* (22.73%), *Actinobacteriota* (11.87%), *Chloroflexi* (10.37%), and *Bacteroidota* (5.73%). The total was 87.68%. *Cyanobacteria* (35.27%), *Actinobacteriota* (17.32%), *Proteobacteria* (13.34%), and *Chloroflexi* (13.07%) were found to have abundances greater than 5% in T9, reaching 79% in total. At the bacterial genus level (Figure 2), only *Microcoleus_PCC-7113* (33.30%) was more abundant than 5% in T7. The genera with abundances greater than 5% in T8 included *Microcoleus_PCC-7113* (35.24%) and *Rheinheimera* (11.63%). The bacterial genera with abundances greater than 5% in T9 included *Microcoleus_PCC-7113* and *Egicoccus* at 28.54% and 5.19%, respectively.

In the beta diversity analysis of the bacterial communities in the three artificial soils (Figure 3), PCA1 and PCA2 in the PcoA (ANOSIM, R = 0.5473, *p* = 0.001) explained 39.95% and 39.26% of the total variation in the bacterial phylum community, respectively, totaling 79.21%. PCA1 and PCA2 in the PcoA (ANOSIM, R = 0.9835, *p* = 0.001) accounted for 46.61% and 29.84% of the total variation in the bacterial genus community, respectively, totaling 76.45%. The results showed that the intra-group variation in bacterial phyla was greater in T8, while it was relatively smaller in T7 and T9. The bacterial phyla and genera among T7~T9 were different, especially at the genus level, and the bacterial community compositions of the three artificial soils were obviously different. As shown in Figure 4, among the three artificial soils, there were significant differences in the abundances of bacterial phyla such as *Proteobacteria*, *Actinobacteriota*, *Firmicutes*, and *Patescibacteria*, and in those of bacterial genera such as *Rheinheimera*, *Egicoccus*, *Actinotalea*, and *Demequina* (*p* < 0.05).

### 3.4. Correlation between Environmental Factors and Bacterial Communities in Artificial Soils

As described in Figure 5, the RDA/CCA of the bacterial communities in the three artificial soils under different environmental factors generally showed that, at the phylum level, the majority of the environmental factors, such as the TN, AN, AP, OM, AK, TP, W_H2O_, S-SC, and S-AP, were positively correlated with the bacterial phyla in T8 to varying degrees but were negatively correlated with the bacterial phyla in T9 to varying degrees. The pH, CEC, S-CAT, S-AKP, and S-UE were positively correlated with the bacterial phyla in T9 but negatively correlated with the bacterial phyla in T8. In addition to the TK, AK, CEC, and S-CAT, the other 11 environmental factors had poor correlations with the bacterial phyla in T7. As shown in Figure 6, among the bacterial phyla with abundances >5%, *Cyanobacteria*, for example, did not significantly correlate with the various environmental factors. *Proteobacteria* was significantly positively correlated with the TN, AN, and AP, while *Actinobacteriota* was significantly negatively correlated with the TN, AN, and AP. *Chloroflexi* was negatively correlated with the AP but positively correlated with the pH and S-AKP. *Bacteroidota* was positively correlated with OM but negatively correlated with S-UE. At the genus level, in general, the bacterial genera in T8 also showed good positive correlations with the W_H2O_, TN, AN, TP, AP, OM, S-SC, and S-AP and good negative correlations with the pH, TK, S-CAT, S-UE, CEC, and S-AKP; T9 and T8 showed opposite characteristics. The bacterial genera in T7 were correlated with the pH, CEC, TK, AK, S-CAT, S-UE, and S-AKP to varying degrees. Among the bacterial genera with abundances >5%, *Microcoleus_PCC-7113*, for example, showed no significant correlation with the various environmental factors. *Rheinheimera* was positively correlated with the TN, AN, AP, and OM but negatively correlated with the pH. *Egicoccus* was positively correlated with the pH, S-UE, and S-AKP but negatively correlated with OM. Moreover, some bacterial phyla with abundances less than 5%, such as *Acidobacteriota*, *Deinococcota*, *Planctomycetota*, and *Desulfobacterota*, as well as some bacterial genera with abundances less than 5%, such as *Demequina*, *Tolypothrix*, *norank_f__JG30-KF-CM45*, *Marmoricola*, and *Arenimonas*, also exhibited significant correlations with the various environmental factors.

## 4. Discussion

### 4.1. Characteristics of Different Environmental Factors in Artificial Soils

The preparation of artificial soil is a highly promising approach for the large-scale, low-cost, and simple collaborative resource utilization of red mud and phosphogypsum [17]. It is expected to be used for ecological reconstruction in tailing areas and for landscaping on municipal roads. However, the research on artificial soil is still in its infancy. Artificial soil is a relatively broad definition of the synergization of red mud and phosphogypsum as the main materials, and whether it has similar fertility characteristics and biological adaptability to natural soil is used as the main evaluation reference. The three artificial soils used in this study have the same material type, but the added amounts are different. Among them, the added amounts of red mud, phosphogypsum, and rice husk are the same, totaling more than 90%, while the added amounts of the other materials, such as bentonite, fly ash, and polyacrylamide flocculant, are small and different, accounting for less than 10%. In general, the three artificial soils are alkaline, and all three can maintain a certain moisture content. According to China’s latest second national soil survey nutrient standard, the three artificial soils have rich contents of TP, TK, AP, AN, and OM, moderate level of CEC, and extremely poor contents of TN and AK, and T8 has better nutrient contents than T7 and T9. The environmental factors of these artificial soils could generally satisfy the growth of most plants. For example, in our previous study [17], the three batches of 12 plants all showed varying degrees of seed germination and seedling growth, and some plants (such as rye grass and *Agrostemma githago*) even grew better than the control group (i.e., the natural soil (NS) group). Moreover, these artificial soils have different enzyme activities, marking their biological (including microbial, plant, animal) activities, which reflect a series of biochemical transformations and the material recycling of various substances in artificial soils, and which also provide evidence of the continuous maturation and evolution of artificial soils [33].

### 4.2. Structure and Function of Bacterial Communities in Artificial Soils

Bacteria can play a pioneering role in all kinds of biochemical reactions and the further development and maturity of soil, and because there are many different kinds of bacteria, compared with plants and animals, they have strong adaptability to different environments and rapid mass reproduction characteristics. Artificial soil already has the characteristics of fertility, which is the basis for the growth of different organisms, but there are also high-salinity and possible heavy metal toxicity risks. The adaptive growth of plants and the settlement of arthropods or other soil animals may take more time, while the strong adaptability, metabolism, growth, and reproduction of bacteria give priority to the rapid colonization and positive transformation of the artificial soil microenvironment. For example, various metabolites produced by bacteria may improve the texture and physicochemical properties of artificial soil [34,35]. The abilities of bacteria, such as carbon and nitrogen fixation and organic matter decomposition, can significantly improve the nutrient level and various enzyme activities of artificial soil, promote nutrient cycling, and further improve the soil fertility [36,37]. A wide variety of bacteria, with their fast growth and reproduction, can continuously enrich the biodiversity of artificial soil. Many bacteria can be symbiotic with plants and animals, which is conducive to enhancing the resistance of plants and animals, allowing them to better adapt to the environment and resist diseases, and to promoting healthy growth, thereby gradually building the ecosystem in the artificial soil and enhancing its ecological function [22,38].

Abundant bacterial species and quantities were formed in the three artificial soils, and there were obvious similarities in the compositions of the bacterial communities (Figure 1 and Figure 2) but also differences to varying degrees (Figure 3 and Figure 4). The same OTUs, phyla, and genera of bacteria in T7~T9 accounted for 58.52%, 82.14%, and 70.07%, respectively, indicating that the three artificial soils had similar environmental conditions. In these artificial soils, the bacterial phyla with high abundances mainly included *Cyanobacteria* (34.79~36.98%), *Proteobacteria* (13.34~22.73%), *Actinobacteriota* (11.87~17.32%), and *Chloroflexi* (10.37~13.07%), which are similar to those in natural soil [39]. These bacterial phyla generally have many beneficial effects on the improvement in the soil quality and fertility, as well as on the resistance to plant growth. In particular, these bacterial phyla also frequently appear in saline–alkali soil and have the functions of protecting plants against saline–alkali stress, normal growth, and soil saline–alkali repair [40,41]. The artificial soils are alkaline and mainly composed of red mud and phosphogypsum with high salinity, so their saline–alkali characteristics are obvious. The colonization of these bacterial communities shows that they are tolerant to salt and alkali, and it is expected to reduce the salinity of artificial soils through acidic metabolite production, among other things, and to have a positive effect on the saline–alkali stress resistance and normal growth of different plants [42]. For example, the bacterium with the highest abundance in the artificial soils was *Cyanobacteria*, which is usually highly adaptable and often appears as a pioneer organism in some unfavorable environments. It can release specific protein signal molecules through photosynthesis to regulate plant growth and development, and it usually has the ability to fix nitrogen [43,44,45,46]. Therefore, it is expected to play an important role in increasing the total nitrogen in artificial soil and in promoting good plant growth. *Proteobacteria* is also a dominant bacterial phylum in artificial soil. Besides playing an important role in improving soil fertility, such as by the decomposition of organic matter and nitrogen fixation, *Proteobacteria* also has a high abundance in some polluted environments and plays a role in environmental pollution remediation, such as in the repair of soil heavy metal pollution, which is very valuable for artificial soil for improving the nutrient circulation and reducing the risk of toxic pollutants [47]. *Actinobacteriota* is one of the main decompositors of plant and animal residues in soil, which can improve the soil nutrient conversion. As a dominant bacterial phylum in artificial soil, *Actinobacteriota* can help improve artificial soil fertility and promote nutrient cycling [48,49,50]. *Chloroflexi* can usually photosynthesize, which is expected to play a beneficial role in artificial soil carbon fixation, the carbon cycle, nitrogen and sulfur conversion, etc. [51,52]. *Bacteroidota* has special functions in polysaccharide decomposition, such as cellulose and carbon and nitrogen metabolism, and rice husk powder in artificial soil contains more cellulose and other polysaccharides, and the artificial soil has high S-SC activity, which is probably related to the strong activity of these bacteria [53,54]. There were many bacterial genera in the three artificial soils, of which *Microcoleus_PCC-7113* had the highest abundance (28.54–33.3%). As a typical photosynthetic bacterium, it has significant functions in carbon fixation, nitrogen fixation, erosion prevention, and biological crust formation in surface soil [45,55]. *Rheinheimera* can degrade organic matter, repair environmental pollution, and reduce the external toxicity of plants [56,57]. *Egicoccus* has strong adaptability to saline–alkali environments and has the function of the ecological restoration of saline–alkali soil [42,58]. The abundance of *Egicoccus* was the highest in T9, which was consistent with the higher pH of T9 compared with that of T7 and T8. In addition, the functional prediction analysis showed that the functions of the bacterial communities in the three artificial soils were very diverse (Figure 7), including photoautotrophy, chemoheterotrophy, aromatic compound degradation, fermentation, nitrate reduction, cellulolysis, and nitrogen fixation, which can significantly affect the artificial soil environment through a very complex variety of bacteria-dominated biochemical reactions and continually promote the improvement in the soil fertility and quality change, while the artificial soil constantly develops to maturity.

### 4.3. Relationship between Bacterial Communities and Environmental Factors in Artificial Soils

Generally, the growth, metabolism, and reproduction of bacteria have a very complex relationship with various soil environmental factors, and the abundances, distributions, and functions of bacterial communities are regulated by a variety of soil environmental factors. In turn, bacteria can continuously enrich the material circulation and energy flow channels of soil through a series of biochemical actions, which can affect the levels of various soil environmental factors [20,21]. In this study, the bacterial communities in the three artificial soils had good correlations with different environmental factors (Figure 5). In particular, some bacterial phyla, such as *Proteobacteria*, *Actinobacteriota*, *Bacteroidota*, and Chloroflexi, and some bacteria genera, such as *Rheinheimera* and *Egicoccus*, had relatively high abundances in the artificial soils and were significantly correlated with major physicochemical factors such as the pH, OM, TN, AN, and AP, and with enzyme activity factors such as S-UE, S-SC, S-AKP, S-CAT, and S-AP. It can be explained that these artificial soil physicochemical factors significantly affect the abundances, distributions, and metabolisms of bacterial communities, and the different enzyme activities in the artificial soils are also largely related to the production of different bacterial biochemical reaction processes. This interactive relationship simultaneously promotes the succession of the bacterial communities and the development of the environmental quality of artificial soils. It should be noted that *Cyanobacteria* and its genus *Microcoleus_PCC-7113*, with the highest abundances in the artificial soils, do not correlate well with various environmental factors, probably because they mainly belong to photoautotrophic bacteria and have low requirements for physicochemical conditions and nutrients in the soil environment, and they can fix nitrogen by assimilating CO_2_ into organic matter and converting N_2_ into ammonia to improve the artificial soil nutrients. The high abundances of these bacteria also indicate that these artificial soils are still in their initial stage [46,59,60]. Moreover, the correlations between the bacterial communities and different environmental factors in T8 and T9 were higher than those in T7, and T8 and T9 exhibited opposite characteristics in general, indicating that they have different environmental quality characteristics and biochemical reaction processes. Some bacterial phyla (*Acidobacteriota*, *Deinococcota*, etc.) and bacterial genera (*Demequina*, *Tolypothrix*, etc.) with low abundances (<5%) were also significantly correlated with many environmental factors. These results comprehensively demonstrate the complexity of the interaction between bacterial communities and artificial soil environmental factors, which may promote the multiple evolution of different artificial soils in terms of these bacterial communities’ succession direction, the environmental quality, the speed of the soil development and maturity, etc. More single-factor regulation experiments and long-term monitoring data are needed for further study.

## 5. Conclusions

We systematically studied the characteristics of the bacterial community structure and function in three artificial soils prepared using red mud and phosphogypsum for the first time. Relatively rich bacterial communities were formed in the artificial soils, with relatively high abundances of several bacteria phyla, such as *Cyanobacteria*, *Proteobacteria*, *Actinobacteriota*, and *Chloroflexi*, and of bacterial genera, such as *Microcoleus_PCC-7113*, *Rheinheimera*, and *Egicoccus*. These bacterial communities play a key role in nutrient transformation, such as carbon fixation, nitrogen fixation, organic decomposition, resistance to saline–alkali stress and pollutant toxicity, the enhancement of various soil enzyme activities, and the ecosystem construction of artificial soils. The functional prediction analysis initially showed that the bacterial communities in the three artificial soils had very diverse functions, including photoautotrophy, chemoheterotrophy, aromatic compound degradation, fermentation, nitrate reduction, cellulolysis, and nitrogen fixation, indicating that these artificial soils have the potential for various biochemical reactions led by bacteria, which can significantly enrich the processes of nutrient cycling and energy flow and enhance the fertility of these artificial soils and the activity of the soil life. The bacterial communities in the different artificial soils were correlated with major physicochemical factors such as the pH, OM, TN, AN, and AP, as well as enzyme activity factors such as S-UE, S-SC, S-AKP, S-CAT, and S-AP, which comprehensively illustrates the complexity of the interaction between the bacterial communities and environmental factors in artificial soils. This may affect the succession direction of different bacterial communities, the quality of the artificial soil environment, and the speed and direction of the development and maturity of artificial soils. This study provides an important scientific basis for the synergistic soilization of two typical industrial solid wastes, that is, red mud and phosphogypsum, for studying the microbial mechanism underlying the evolution and development of artificial soils prepared using these wastes.

## Figures and Tables

**Figure 1 microorganisms-12-01886-f001:**
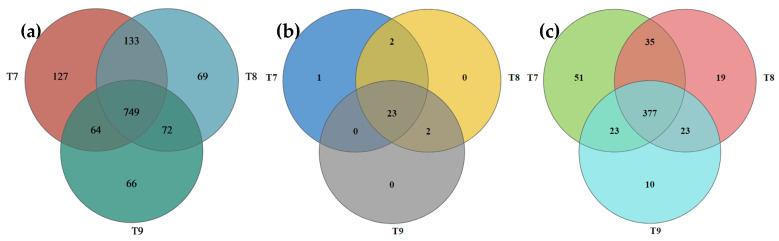
Venn analysis of bacterial communities (OTUs (**a**), phyla (**b**), and genera (**c**)) in artificial soils.

**Figure 2 microorganisms-12-01886-f002:**
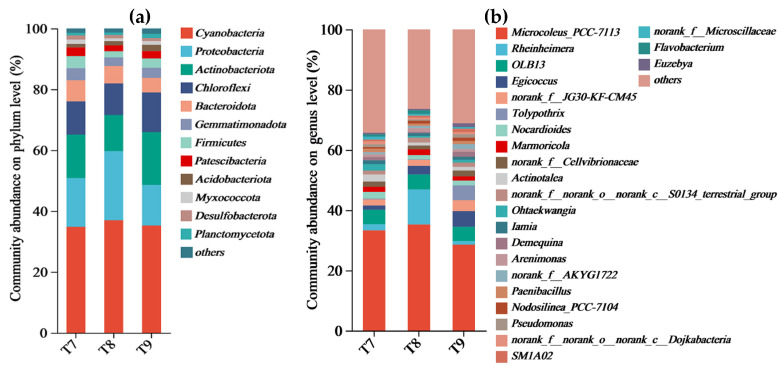
Composition of bacterial communities (at phylum level (**a**) and genus level (**b**)) in artificial soils.

**Figure 3 microorganisms-12-01886-f003:**
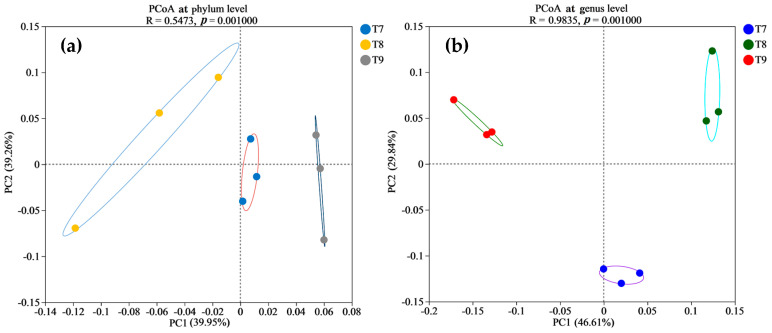
PCoA of beta diversity of bacterial communities (at phylum level (**a**) and genus level (**b**)) in artificial soils.

**Figure 4 microorganisms-12-01886-f004:**
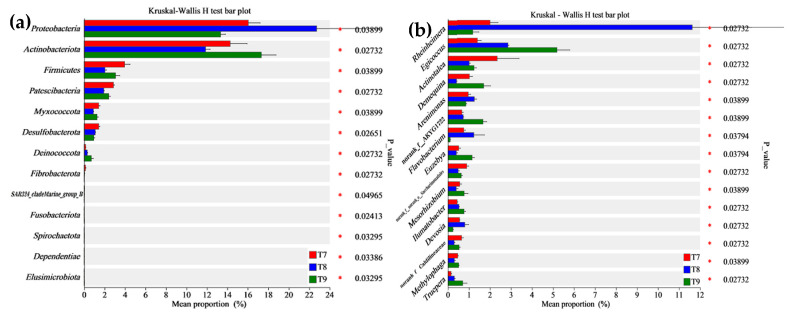
Kruskal–Wallis H test analysis of bacterial communities (phyla (**a**) and genera (**b**)) in artificial soils.

**Figure 5 microorganisms-12-01886-f005:**
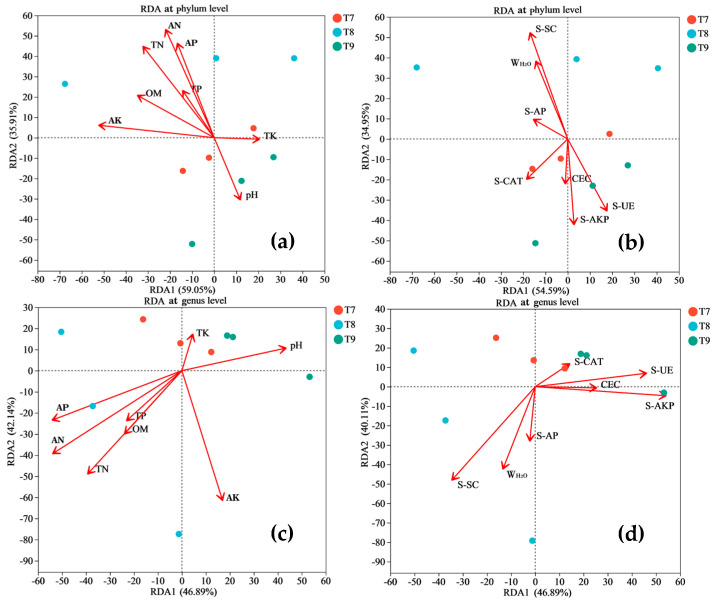
RDA/CCA of bacterial communities (at phylum level (**a**,**b**) and genus level (**c**,**d**)) in artificial soils.

**Figure 6 microorganisms-12-01886-f006:**
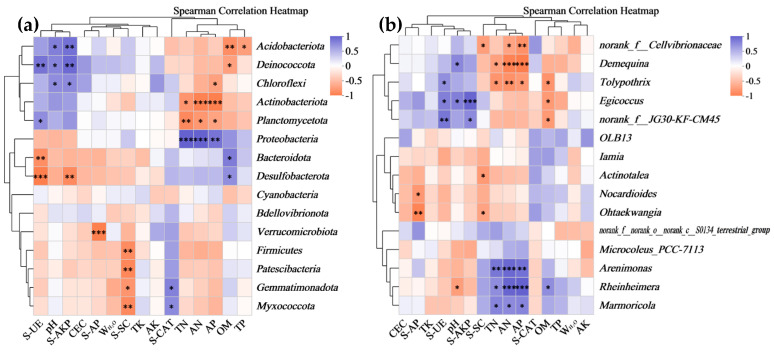
Spearman correlation heatmap analysis of bacterial communities (phyla (**a**) and genera (**b**)) in artificial soils. Note: * 0.01 < *p* ≤ 0.05; ** 0.001< *p* ≤ 0.01; *** *p* ≤ 0.001.

**Figure 7 microorganisms-12-01886-f007:**
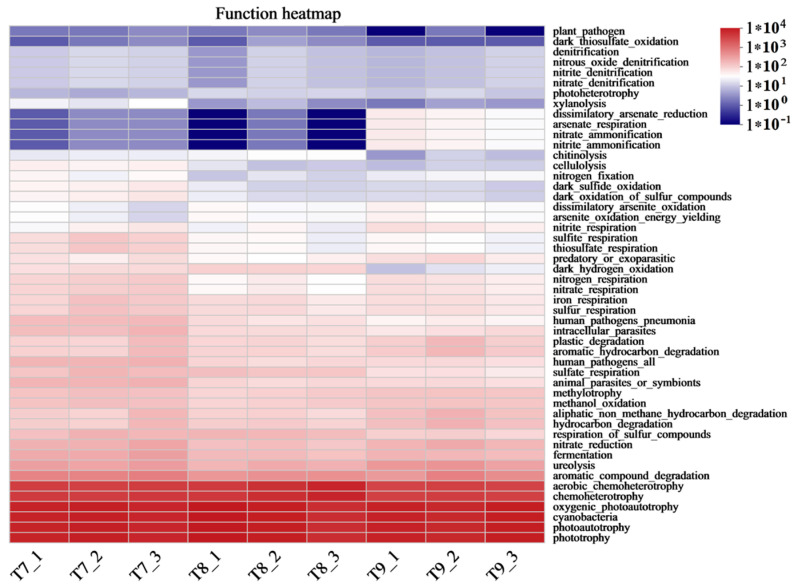
FAPROTAX functional prediction analysis of bacterial communities in artificial soils.

**Table 1 microorganisms-12-01886-t001:** Compositions of artificial soils prepared using red mud and phosphogypsum.

Name	Red Mud–Phosphogypsum Substrate (g)	Rice Hull Powder (g)	Bentonite (g)	Fly Ash (g)	Polyacrylamide Flocculant (g)
T7	200	20	4	10	0.5
T8	200	20	10	2	1
T9	200	20	20	5	0.25

**Table 2 microorganisms-12-01886-t002:** Common physicochemical parameters of artificial soils.

Name	pH	W_H2O_ (%)	OM (%)	CEC (cmol/kg)
T7	8.4 ± 0.1 b	30.1 ± 1.5 a	5.5 ± 0.1 a	11.1 ± 0.3 a
T8	8.4 ± 0.0 b	31.7 ± 2.1 a	5.4 ± 0.3 a	11.4 ± 0.4 a
T9	8.7 ± 0.1 a	30.9 ± 1.8 a	5.0 ± 0.1 b	12.3 ± 1.5 a

Note: pH represents pH_H2O_; W_H2O_: moisture content; OM: organic matter; CEC: cation-exchange capacity. The different letters indicate that there is a significant difference in the same parameter among T7~T9 (*p* < 0.05).

**Table 3 microorganisms-12-01886-t003:** Main nutrients of artificial soils.

Name	TN (mg/kg)	AN (mg/kg)	TP (mg/kg)	AP (mg/kg)	TK (g/kg)	AK (mg/kg)
T7	445.5 ± 52.2 b	153.6 ± 22.0 b	2726.4 ± 196.8 a	482.7 ± 42.4 b	31.8 ± 1.8 a	14.2 ± 0.6 a
T8	596.0 ± 83.8 a	211.4 ± 8.2 a	2842.9 ± 454.7 a	548.2 ± 19.5 a	32.4 ± 4.4 a	15.4 ± 4.3 a
T9	395.2 ± 80.4 b	110.5 ± 8.3 c	2578.6 ± 108.3 a	180.8 ± 31.7 c	38.6 ± 11.9 a	14.5 ± 0.9 a

Note: TN: total nitrogen; TP: total phosphorous; TK: total potassium; AN: available nitrogen; AP: available phosphorous; AK: available potassium. The different letters indicate that there is a significant difference in the same parameter among T7~T9 (*p* < 0.05).

**Table 4 microorganisms-12-01886-t004:** Activity of common enzymes of artificial soils.

Name	S-CAT (ml/g)	S-AKP (mg/g, 24 h)	S-UE (mg/g, 24 h)	S-AP (mg/g, 24 h)	S-SC (mg/g, 24 h)
T7	1.285 ± 0.071 a	0.329 ± 0.049 c	0.318 ± 0.058 b	0.028 ± 0.002 a	1.020 ± 0.238 b
T8	1.121 ± 0.121 a	0.417 ± 0.039 b	0.425 ± 0.072 b	0.030 ± 0.002 a	1.762 ± 0.385 a
T9	1.162 ± 0.073 a	0.561 ± 0.004 a	0.789 ± 0.176 a	0.029 ± 0.004 a	1.253 ± 0.008 b

Note: S-CAT: catalase; S-AKP: alkaline phosphatase; S-UE: urease; S-AP: acid phosphatase; S-SC: sucrase. The different letters indicate that there is a significant difference in the same parameter among T7~T9 (*p* < 0.05).

**Table 5 microorganisms-12-01886-t005:** Alpha diversity indexes of bacterial communities in artificial soils.

Name	Sobs	Shannon	Simpson	Ace	Chao	Pielou_e	Coverage
T7-Mean	861.67 a	4.16 a	0.12 a	1026.61 a	1038.6 7a	0.61 a	0.9935 a
T7-Stdev	11.72	0.166	0.02	31.97	57.85	0.02	0.0005
T8-Mean	825.00 b	3.706 b	0.15 a	983.437 ab	980.85 ab	0.55 b	0.9937 a
T8-Stdev	17.78	0.176	0.05	44.73	57.27	0.03	0.0007
T9-Mean	768.67 c	4.116 a	0.09 a	913.94 b	926.07 b	0.62 a	0.9942 a
T9-Stdev	15.14	0.206	0.02	26.01	25.86	0.03	0.0002

Note: The different letters indicate that there is a significant difference in the same parameter among T7~T9 (*p* < 0.05).

## Data Availability

The original contributions presented in the study are included in the article, further inquiries can be directed to the corresponding author.

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
