# Peer review of "Characteristics of Bacterial Community Structure and Function in Artificial Soil Prepared Using Red Mud and Phosphogypsum"

_microorganisms, 2024, doi:10.3390/microorganisms12091886_

Round 1

Reviewer 1 Report

Comments and Suggestions for Authors

It is a very interesting topic and this manuscript is an important contribution to the research community in this area.

Abstract part provides o good and comprehensive overview of the topic and background.

In section Materials and methods line 123- Data on how manyand which ones exactly microorganisms were in 10 ml are missing. After full mixing, 10 ml of dilute suspension of microorganisms (Mainly bacillus, such as Bacillus amyloliquefaciens, Bacillus licheniformis, Bacillus  subtilis, etc.). Was there a control ie. variants without the addition of microorganisms?

What plants were grown on that artificial soil? Soils without inoculation - T7, T8, T9, should be compared  with inoculated soils.

Line 147 -main nutrients (TN, TP, TK, AN, AP, AK), the full name should be added - Total nitrogen (TN) and for all.

Author Response

Dear reviewer,

Thank you very much for your review, which provides good suggestions. My response as follows,

Reviewer :

It is a very interesting topic and this manuscript is an important contribution to the research community in this area. Abstract part provides o good and comprehensive overview of the topic and background.

In section Materials and methods line 123- Data on how many and which ones exactly microorganisms were in 10 ml are missing. After full mixing, 10 ml of dilute suspension of microorganisms (Mainly bacillus, such as Bacillus amyloliquefaciens, Bacillus licheniformis, Bacillus  subtilis, etc.). Was there a control ie. variants without the addition of microorganisms? What plants were grown on that artificial soil? Soils without inoculation - T7, T8, T9, should be compared  with inoculated soils. Line 147 -main nutrients (TN, TP, TK, AN, AP, AK), the full name should be added - Total nitrogen (TN) and for all.

Response: Thank you very much for your review, which provides good suggestions. We have modified the concentration of dilute suspension of microorganisms, please see the “2.1. Preparation of artificial soil” of “Revised manuscript” for details. The purpose of our study is to analyze the composition of artificial soil bacterial community and its functional characteristics. 10 ml dilute suspension of microorganisms were added into each of the three artificial soils and as an initial source of soil microbes. We did not set up a control group without adding microorganisms, because we believe that some well-adapted microbial communities will survive and reproduce gradually according to the soil environment during artificial soil development stage, so a control group is not essential. Of course, if there is a control group, it may be possible to have a more comprehensive understanding of the microbial formation process in artificial soil.

Three batches of plant (a total of 12 plants, such as rapeseed, chili pepper, alfalfa, and rye grass, etc.) were grown. Because this study is related to our previous research, we mainly described this information in the form of annotated reference in the manuscript, plese see the reference 17 of “Revised manuscript” and the “2.2. Sample collection and determination”of “Revised manuscript” for details (only minor changes have been made). Through initial microbial inoculation and plant cultivation, we provide an initial source of microorganisms and verify plant growth as well as promote the continuous evolution of artificial soils, respectively.

We have modified the full name of main nutrients (TN, TP, TK, AN, AP, AK), please see the “2.2. Sample collection and determination” of “Revised manuscript” for details.

If there is anything that needs to be further modified, we hope to have the opportunity to modify it again. Thank you very much.

With Best Regards

Dr. Liu

Reviewer 2 Report

Comments and Suggestions for Authors

The manuscript submitted for review addresses a very important aspect of the management of post-industrial waste on the one hand and the problem of soil fertility on the other. The proposed approach of using two important wastes - red mud and phosphogypsum - and composing artificial soils is an interesting approach. After composing the soils, the authors introduced mixtures of selected bacteria and cultivated the plants for one year. However, in the paper, the authors do not show if or how both the soil parameters and the enzymatic activity of these mixtures of microorganisms change during the course of the experiment (even if only at the beginning and end of the experiment). What is the dynamism of the systems obtained. The work should be detailed so that this information is clear to the reader.

In addition, the manuscript lacks some information. I attach my detailed comments below

1. why was the total mass of the composed soils different? (table 1)

2. The methodology  should be supplemented by a description of the tests for physico-chemical parameters of the soils?

3. In methodology is also no information on what and how the enzyme activity of the soils was checked?

 4. regarding section 2.2. - The authors mention that they cultivated plants in pots for one year? There is no information on what size pots were made of what material, how much soil was placed? Were they tested under controlled conditions or not? Were photoperiod and temperature controls used?  What was used to irrigate and keep the humidity constant?

Furthermore, the authors do not state what plants were used in the study?

If studies on plants have already been published, it should be clearly stated and the literature reference should be given.

Author Response

Dear reviewer,

Thank you very much for your review, which provides good suggestions. My response as follows,

Reviewer:

The manuscript submitted for review addresses a very important aspect of the management of post-industrial waste on the one hand and the problem of soil fertility on the other. The proposed approach of using two important wastes - red mud and phosphogypsum - and composing artificial soils is an interesting approach. After composing the soils, the authors introduced mixtures of selected bacteria and cultivated the plants for one year. However, in the paper, the authors do not show if or how both the soil parameters and the enzymatic activity of these mixtures of microorganisms change during the course of the experiment (even if only at the beginning and end of the experiment). What is the dynamism of the systems obtained. The work should be detailed so that this information is clear to the reader. In addition, the manuscript lacks some information. I attach my detailed comments below

Response: Thank you very much for your review, which provides good suggestions. This study is related to our previous research, some informations were described in the form of annotated reference in the manuscript, plese see the reference 17 of “Revised manuscript” for details. The study on the preparation of artificial soil using red mud and phosphogypgypsum is still in its infancy, we analyze the composition of artificial soil bacterial community and its functional characte for the first time, which provides an important scientific basis for further understanding of the microbiological mechanism of the further evolution and development of red mud and phosphogypsum based artificial soil. The physical and chemical characteristics of artificial soil, nutrient level, plant planting, cultivation time, etc. can all be used as driving factors to affect microbial community diversity. Therefore, the dynamic change of microbial community diversity in artificial soil and its driving mechanism are very important topics, which need to set up more systematic single-factor experiments and monitoring at different stages, and belong to one of the future research directions.

  1. why was the total mass of the composed soils different? (table 1)

Response: Thank you very much for your review, which provides good suggestions for the optimization of experimental design. The total mass of the three artificial soils in Table 1 are different, which is really confusing. However, it should be noted that our research idea is to first determine the mass ratio of neutralization reaction between red mud and phosphogypsum through experiments. Based on the neutral red mud-phosphogypsum substrate, then some other auxiliary materials are added to optimize the neutral substrate for forming the artificial soil with fertility characteristics. The total mass of artificial soil will be different due to the different amount of auxiliary materials added, we believe that this is a reasonable experimental design scheme that will help us formulate artificial soils with different material ratios for more research.

  1. The methodology  should be supplemented by a description of the tests for physico-chemical parameters of the soils?

Response: Thank you very much for your review, which provides good suggestions. We have modified the methodology, plese see the “2.2. Sample collection and determination”of “Revised manuscript” for details.

  1. In methodology is also no information on what and how the enzyme activity of the soils was checked?

Response: Thank you very much for your review, which provides good suggestions. We have modified the methodology, plese see the “2.2. Sample collection and determination” of “Revised manuscript” for details.

  1. regarding section 2.2. - The authors mention that they cultivated plants in pots for one year? There is no information on what size pots were made of what material, how much soil was placed? Were they tested under controlled conditions or not? Were photoperiod and temperature controls used?  What was used to irrigate and keep the humidity constant? Furthermore, the authors do not state what plants were used in the study? If studies on plants have already been published, it should be clearly stated and the literature reference should be given.

Response: Thank you very much for your review, which provides good suggestions. Because this study is related to our previous research, some informations were described in the form of annotated reference in the manuscript, plese see the reference 17 of “Revised manuscript” for details. We placed artificial soil in each pot according to the ratio and total mass in Table 1. Three batches of plant (a total of 12 plants, such as rapeseed, chili pepper, alfalfa, and rye grass, etc.) were grown as described in the reference 17 of “Revised manuscript” for details. Regular watering, so that the artificial soil always maintain a moisture content of about 30%, and the plant pot experiment is carried out in a greenhouse at a temperature of about 25~30℃. We have modified some informations in the “2.2. Sample collection and determination” section of “Revised manuscript”

If there is anything that needs to be further modified, we hope to have the opportunity to modify it again. Thank you very much.

With Best Regards

Dr. Liu

Reviewer 3 Report

Comments and Suggestions for Authors

Comments and suggestions for Authors

Title: Characteristics of Bacterial Community Structure and Function in
Artificial Soil Prepared by Red Mud and Phosphogypsum

The manuscript is interesting and the content it contains fits the publishing profile of Microorganisms journal. The results were clearly presented and described. The discussion of the results is generally well written. The research conducted on a mixture of waste materials is ultimately bold and, however, poses a threat to the protection of the soil environment. Similar studies could be considered with the addition of red mud and phosphogypsum to traditional soil.

Remarks:

In order to improve the manuscript, authors should address the following comments.

§  Materials and Methods: In my opinion, this is not traditional soil and not even (artificial). It is a prepared mixture of waste materials. In this section, please provide the chemical composition and origin of the main and auxiliary starting materials. Subsection 2.2. The principles of the methods for determining the listed parameters should be given. Seeds of what plants were used for research?

§  Results and analysis: A subsection on seed germination and plant growth should be added. Table 1 - You need to add pHKCl or pH H2O. Explanations of abbreviations and statistical references should be included below the tables.

§  Discussion: Lines 285-287 On what basis did the Authors determine that the three types of waste material mixtures are weakly alkaline or more alkaline (pH 8.4-8.7)?

§  Conclusion: Lines 414-420 Functions of bacterial communities that were not studied by the authors should be removed.

§  References should be adapted to editorial requirements.

Best regards

Author Response

Dear reviewer,

Thank you very much for your review, which provides good suggestions. My response as follows,

Reviewer:

The manuscript is interesting and the content it contains fits the publishing profile of Microorganisms journal. The results were clearly presented and described. The discussion of the results is generally well written. The research conducted on a mixture of waste materials is ultimately bold and, however, poses a threat to the protection of the soil environment. Similar studies could be considered with the addition of red mud and phosphogypsum to traditional soil.  Remarks: In order to improve the manuscript, authors should address the following comments.

  • Materials and Methods: In my opinion, this is not traditional soil and not even (artificial). It is a prepared mixture of waste materials. In this section, please provide the chemical composition and origin of the main and auxiliary starting materials. Subsection 2.2. The principles of the methods for determining the listed parameters should be given. Seeds of what plants were used for research?

Response: Thank you very much for your review, which provides very good suggestions for us to think about how to more accurately define this mixed material that can be used for vegetation restoration. We consider that the mixed material is mainly prepared by artificially regulating the neutralization reaction of red mud and phosphogypsum (accounting for more than 85%), and adding a small amount of other auxiliary materials. This mixed material has fertility and soil-like properties, it can be used for plant restoration, so we call it artificial soil. Such studies on the preparation of artificial soil with red mud and phosphogypsum as the main materials are still almost blank. Our previous research have systematically verified the huge potential of artificial soil for vegetation restoration through the analysis of physical and chemical characteristics, main nutrients and plant growth, etc. In this study, we focus on the systematic analysis of the bacterial community structure and the potential functional characteristics of the main bacteria in the artificial soil based on our previous research, in order to provide reference for the microbial mechanism in the development and evolution of artificial soil. Therefore, information about the source and main chemical components of red mud, phosphogypsum and other materials has been described in our previous research, and these materials are relatively common and have been reported in many other literatures, also, the plant pot tests and related index testing methods have also been detailed in our previous research, so the description in this study is relatively limited, mainly in the form of reference remarks, and we have made some revisions as far as possible, please see the “2. Materials and methods” of “Revised manuscript” for details.

  • Results and analysis: A subsection on seed germination and plant growth should be added. Table 1 - You need to add pHKCl or pH H2O. Explanations of abbreviations and statistical references should be included below the tables.

Response: Thank you very much for your review, which provides very good suggestions. The plant pot test and related index testing methods have been detailed in our previous research, and the systematic analysis of the bacterial community structure and the potential functional characteristics of the main bacteria in the artificial soil is emphasized in this study. We considered that the introduction of seed germination and plant growth by a separate subsection in this study is repetitive, so the description is relatively limited, mainly in the form of reference remarks, and we have made some revisions in the “4. Discussion, 4.1. Characteristics of different environmental factors in artificial soils” of “Revised manuscript” for details. As for the clear meaning of pH, explanations of abbreviations and statistical references in the tables, we have repeatedly checked and made obvious modifications, please see the tables of “Revised manuscript” for details.

  • Discussion:Lines 285-287 On what basis did the Authors determine that the three types of waste material mixtures are weakly alkaline or more alkaline (pH 8.4-8.7)?

Response: Thank you very much for your review, which provides very good suggestions. We have made some modifications, and the expression is more rigorous than before, that is, all three artificial soils are alkaline. Please see the “4. Discussion, 4.1. Characteristics of different environmental factors in artificial soils” of “Revised manuscript” for details.

  • Conclusion:Lines 414-420 Functions of bacterial communities that were not studied by the authors should be removed.

Response: Thank you very much for your review, which provides very good suggestions. In this study, the functional analysis of bacterial community in artificial soil was not comprehensive and in-depth enough. However, we carried out the functional prediction analysis of bacterial community, as shown in Figure 7, and discussed related issues. This preliminary result can help to understand which key biochemical reactions can be led by bacteria community in artificial soil to promote the further maturation of artificial soil, so we tend to keep that conclusion (only minor changes have been made). If you think it is still necessary to remove, please give us the opportunity to further modify, thank you very much.

  • References should be adapted to editorial requirements.

Response: Thank you very much for your review, which provides very good suggestions. We have modified them.

If there is anything that needs to be further modified, we hope to have the opportunity to modify it again. Thank you very much.

With Best Regards

Dr. Liu

Round 2

Reviewer 2 Report

Comments and Suggestions for Authors

I have no comments on the revised and corrected manuscript .